# A Reduced-Dimension fMRI Shared Response Model

**Po-Hsuan Chen**[1]**, Janice Chen**[2]**, Yaara Yeshurun**[2]**,**
**Uri Hasson**[2]**, James V. Haxby**[3]**, Peter J. Ramadge**[1]
[1]Department of Electrical Engineering, Princeton University
[2]Princeton Neuroscience Institute and Department of Psychology, Princeton University
[3]Department of Psychological and Brain Sciences
and Center for Cognitive Neuroscience, Dartmouth College

## Abstract

Multi-subject fMRI data is critical for evaluating the generality and validity of findings across subjects, and its effective utilization helps improve analysis sensitivity. We develop a shared response model for aggregating multi-subject fMRI data that accounts for different functional topographies among anatomically aligned datasets. Our model demonstrates improved sensitivity in identifying a shared response for a variety of datasets and anatomical brain regions of interest. Furthermore, by removing the identified shared response, it allows improved detection of group differences. The ability to identify what is shared and what is not shared opens the model to a wide range of multi-subject fMRI studies.

## 1 Introduction

Many modern fMRI studies of the human brain use data from multiple subjects. The use of multiple subjects is critical for assessing the generality and validity of the findings across subjects. It is also increasingly important since from one subject one can gather at most a few thousand noisy instances of functional response patterns. To increase the power of multivariate statistical analysis, one therefore needs to aggregate response data across multiple subjects. However, the successful aggregation of fMRI brain imaging data across subjects requires resolving the major problem that both anatomical structure and functional topography vary across subjects [1, 2, 3, 4]. Moreover, it is well known that standard methods of anatomical alignment [1, 4, 5] do not adequately align functional topography [4, 6, 7, 8, 9]. Hence anatomical alignment is often followed by spatial smoothing of the data to blur functional topographies. Recently, functional spatial registration methods have appeared that use cortical warping to maximize inter-subject correlation of time series [7] or inter-subject correlation of functional connectivity [8, 9]. A more radical approach learns a latent multivariate feature that models the shared component of each subject's response [10, 11, 12].

Multivariate statistical analysis often begins by identifying a set of features that capture the informative aspects of the data. For example, in fMRI analysis one might select a subset of voxels within an anatomical region of interest (ROI), or select a subset of principal components of the ROI, then use these features for subsequent analysis. In a similar way, one can think of the fMRI data aggregation problem as a two step process. First use training data to learn a mapping of each subject's measured data to a shared feature space in a way that captures the across-subject shared response. Then use these learned mappings to project held out data for each subject into the shared feature space and perform a statistical analysis.

To make this more precise, let $\{X_i \in \mathbb{R}^{v \times d}\}_{i=1}^{m}$ denote matrices of training data ($v$ voxels in the ROI, over $d$ TRs) for $m$ subjects. We propose using this data to learn subject specific bases $W_i \in \mathbb{R}^{v \times k}$, where $k$ is to be selected, and a shared matrix $S \in \mathbb{R}^{k \times d}$ of feature responses such that $X_i = W_i S + E_i$ where $E_i$ is an error term corresponding to unmodeled aspects of the subject's response. One can think of the bases $W_i$ as representing the individual functional topographies and $S$ as a latent feature that captures the component of the response shared across subjects. We don't claim that $S$ is a sufficient statistic, but that is a useful analogy.

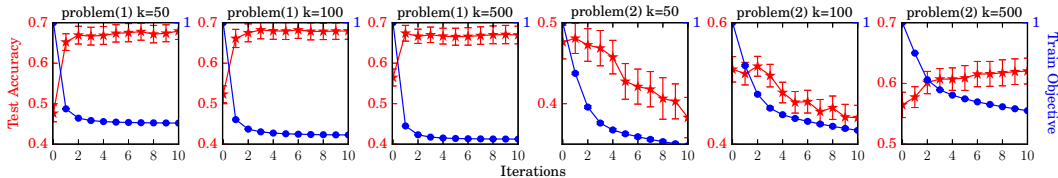

Figure 1: Comparison of training objective value and testing accuracy for problem (1) and (2) over various $k$ on *raider* dataset with 500 voxels of ventral temporal cortex (VT) in image stimulus classficiation experiment (details in Sec.4). In all cases, error bars show $\pm 1$ standard error.

The contribution of the paper is twofold: First, we propose a probabilistic generative framework for modeling and estimating the subject specific bases $W_i$ and the shared response latent variable $S$. A critical aspect of the model is that it directly estimates $k \ll v$ shared features. This is in contrast to methods where the number of features equals the number of voxels [10, 11]. Moreover, the Bayesian nature of the approach provides a natural means of incorporating prior domain knowledge. Second, we give a demonstration of the robustness and effectiveness of our data aggregation model using a variety of fMRI datasets captured on different MRI machines, employing distinct analysis pathways, and based on various brain ROIs.

## 2 Preliminaries

fMRI time-series data $X_i \in \mathbb{R}^{v \times d}$, $i = 1{:}m$, is collected for $m$ subjects as they are presented with identical, time synchronized stimuli. Here $d$ is the number of time samples in TRs (Time of Repetition), and $v$ is the number of voxels. Our objective is to model each subject's response as $X_i = W_i S + E_i$ where $W_i \in \mathbb{R}^{v \times k}$ is a basis of topographies for subject $i$, $k$ is a parameter selected by the experimenter, $S \in \mathbb{R}^{k \times d}$ is a corresponding time series of shared response coordinates, and $E_i$ is an error term, $i = 1{:}m$. To ensure uniqueness of coordinates it is necessary that $W_i$ has linearly independent columns. We make the stronger assumption that each $W_i$ has orthonormal columns, $W_i^T W_i = I_k$.

Two approaches for estimating the bases $W_i$ and the shared response $S$ are illustrated below:

$$\begin{aligned} \min_{W_i,S} \quad & \textstyle\sum_i \|X_i - W_i S\|_F^2 \\ \text{s.t.} \quad & W_i^T W_i = I_k, \end{aligned} \tag{1}$$

$$\begin{aligned} \min_{W_i,S} \quad & \textstyle\sum_i \|W_i^T X_i - S\|_F^2 \\ \text{s.t.} \quad & W_i^T W_i = I_k, \end{aligned} \tag{2}$$

where $\|\cdot\|_F$ denotes the Frobenius norm. For $k \leq v$, (1) can be solved iteratively by first selecting initial conditions for $W_i$, $i = 1{:}m$, and optimizing (1) with respect to $S$ by setting $S = 1/m \sum_i W_i^T X_i$. With $S$ fixed, (1) becomes $m$ separate subproblems of the form $\min \|X_i - W_i S\|_F^2$ with solution $W_i = \tilde{U}_i \tilde{V}_i^T$, where $\tilde{U}_i \tilde{\Sigma}_i \tilde{V}_i^T$ is an SVD of $X_i S^T$ [13]. These two steps can be iterated until a stopping criterion is satisfied. Similarly, for $k \leq v$, (2) can also be solved iteratively. However, for $k < v$, there is no known fast update of $W_i$ given $S$. Hence this must be done using local gradient decent on the Stiefel manifold [14]. Both approaches yield the same solution when $k = v$, but are not equivalent in the more interesting situation $k \ll v$ (Sup. Mat.). What is most important, however, is that problem (2) with $k < v$, often learns an uninformative shared response $S$. This is illustrated in Fig. 1 which plots of the value of the training objective and the test accuracy for a stimulus classification experiment versus iteration count (image classification using the *raider* fMRI dataset, see Sec.4). For problem (1), test accuracy increases with decreasing training error, Whereas for problem (2), test accuracy decreases with decreasing training error (This can be explained analytically, see Sup. Mat.). We therefore base our approach on a generalization of problem (1). We call the resulting $S$ and $\{W_i\}_{i=1}^m$ a shared response model (SRM).

Before extending this simple model, we note a few important properties. First, a solution of (1) is not unique. If $S$, $\{W_i\}_{i=1}^m$ is a solution, then so is $QS$, $\{W_i Q^T\}_{i=1}^m$, for any $k \times k$ orthogonal matrix $Q$. This is not a problem as long as we only learn one template and one set of subject bases. Any new subjects or new data will be referenced to the original SRM. However, if we independently learn two SRMs, the group shared responses $S_1, S_2$, may not be registered (use the same $Q$). We register $S_1$ to $S_2$ by finding a $k \times k$ orthogonal matrix $Q$ to minimize $\|S_2 - QS_1\|_F^2$; then use $QS_1$ in place of $S_1$ and $W_j Q^T$ in place of $W_j$ for subjects in the first SRM.

Next, when projected onto the span of its basis, each subject's training data $X_i$ has coordinates $S_i = W_i^T X_i$ and the learning phase ensures $S = 1/m \sum_i^m S_i$. The projection to $k$ shared features

and the averaging across subjects in *feature space* both contribute to across-subject denoising during the learning phase. By mapping $S$ back into voxel space we obtain the voxel space manifestation $W_i S$ of the denoised, shared component of each subject's training data. The training data of subject $j$ can also be mapped through the shared response model to the functional topography and anatomy of subject $i$ by the mapping $\hat{X}_{i,j} = W_i W_j^T X_j$.

New subjects are easily added to an existing SRM $S$, $\{W_i\}_{i=1}^m$. We refer to $S$ as the training template. To introduce a new subject $j = m + 1$ with training data $X_j$, form its orthonormal basis by minimizing the mean squared modeling error $\min_{W_j, W_j^T W_j = I_k} \|X_j - W_j S\|_F^2$. We solve this for the least norm solution. Note that $S$, and the existing $W_{1:m}$ do not change; we simply add a new subject by using its training data for the same stimulus and the template $S$ to determine its basis of functional topographies. We can also add new data to an SRM. Let $X_i'$, $i = 1{:}m$, denote new data collected under a distinct stimulus from the same subjects. This is added to the study by forming $S_i' = W_i^T X_i'$, then averaging these projections to form the shared response for the new data: $S' = \frac{1}{m} \sum_{i=1}^m W_i^T X_i'$. This assumes the learned subject specific topographies $W_i$ generalize to the new data. This usually requires a sufficiently rich stimulus in the learning phase.

## 3 Probabilistic Shared Response Model

We now extend our simple shared response model to a probabilistic setting. Let $x_{it} \in \mathbb{R}^v$ denote the observed pattern of voxel responses of the $i$-th subject at time $t$. For the moment, assume these observations are centered over time. Let $s_t \in \mathbb{R}^k$ be a hyperparameter modeling the shared response at time $t = 1{:}d$, and model the observation at time $t$ for dataset $i$ as the outcome of a random vector:

$$\mathbf{x}_{it} \sim \mathcal{N}(W_i s_t, \rho^2 I), \qquad \text{with} \quad W_i^T W_i = I_k, \tag{3}$$

where, $\mathbf{x}_{it}$ takes values in $\mathbb{R}^v$, $W_i \in \mathbb{R}^{v \times k}$, $i = 1{:}m$, and $\rho^2$ is a subject independent hyperparameter. The negative log-likelihood of this model is $\mathcal{L} = \sum_t \sum_i \frac{v}{2} \log 2\pi + \frac{v}{2} \log \rho^2 + \frac{\rho^{-2}}{2}(x_{it} - W_i s_t)^T (x_{it} - W_i s_t)$. Noting that $x_{it}$ is the $t$-th column of $X_i$, we see that minimizing $\mathcal{L}$ with respect to $W_i$ and $S = [s_1, \ldots, s_d]$, requires the solution of:

$$\min \sum_t \sum_i (x_{it} - W_i s_t)^T (x_{it} - W_i s_t) = \min \sum_i \|X_i - W_i S\|_F^2.$$

Thus maximum likelihood estimation for this model matches (1).

In our fMRI datasets, and most multi-subject fMRI datasets available today, $d \gg m$. Since $\mathbf{s}_t$ is time specific but shared across the $m$ subjects, we see that there is palpable value in regularizing its estimation. In contrast, subject specific variables such as $W_i$ are shared across time, a dimension in which data is relatively plentiful. Hence, a natural extension of (3) is to make $s_t$ a shared latent random vector $\mathbf{s}_t \sim \mathcal{N}(\mathbf{0}, \Sigma_s)$ taking values in $\mathbb{R}^k$. The observation for dataset $i$ at time $t$ then has the conditional density $p(\mathbf{x}_{it} | \mathbf{s}_t) = \mathcal{N}(W_i \mathbf{s}_t + \mu_i, \rho_i^2 I)$, where the subject specific mean $\mu_i$ allows for a non-zero mean and we assume subject dependent isotropic noise covariance $\rho_i^2 I$. This is an extended multi-subject form of factor analysis, but in factor analysis one normally assumes $\Sigma_s = I$.

To form a joint model, let $\mathbf{x}_t^T = [\mathbf{x}_{1t}^T \ldots \mathbf{x}_{mt}^T]$, $W^T = [W_1^T \ldots W_m^T]$, $\mu^T = [\mu_1^T \ldots \mu_m^T]$, $\Psi = \text{diag}(\rho_1^2 I, \ldots, \rho_m^2 I)$, $\boldsymbol{\epsilon} \sim \mathcal{N}(0, \Psi)$, and $\Sigma_x = W \Sigma_s W^T + \Psi$. Then

$$\mathbf{x}_t = W \mathbf{s}_t + \mu + \boldsymbol{\epsilon}, \tag{4}$$

with $\mathbf{x}_t \sim \mathcal{N}(\mu, \Sigma_x)$ taking values in $\mathbb{R}^{mv}$. For this joint model, we formulate SRM as:

$$\begin{aligned} \mathbf{s}_t &\sim \mathcal{N}(0, \Sigma_s), \\ \mathbf{x}_{it} | \mathbf{s}_t &\sim \mathcal{N}(W_i \mathbf{s}_t + \mu_i, \rho_i^2 I), \\ W_i^T W_i &= I_k, \end{aligned} \tag{5}$$

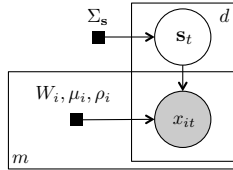

Figure 2: Graphical model for SRM. Shaded nodes: observations, unshaded nodes: latent variables, and black squares: hyperparameters.

where $\mathbf{s}_t$ takes values in $\mathbb{R}^k$, $\mathbf{x}_{it}$ takes values in $\mathbb{R}^v$, and the hyperparameters $W_i$ are matrices in $\mathbb{R}^{v \times k}$, $i = 1{:}m$. The latent variable $\mathbf{s}_t$, with covariance $\Sigma_s$, models a shared elicited response across the subjects at time $t$. By applying the same orthogonal transform to each of the $W_i$, we can assume, without loss of generality, that $\Sigma_s$ is diagonal. The SRM graphical model is displayed in Fig. 2.

## 3.1 Parameter Estimation for SRM

To estimate the parameters of the SRM model we apply a constrained EM algorithm to find maximum likelihood solutions. Let $\theta$ denote the vector of all parameters. In the E-step, given initial value or estimated value $\theta^{\text{old}}$ from the previous M-step, we calculate the sufficient statistics by taking expectation with respect to $p(\mathbf{s}_t|x_t, \theta^{\text{old}})$:

$$\mathbb{E}_{\mathbf{s}|x}[\mathbf{s}_t] = (W\Sigma_s)^T (W\Sigma_s W^T + \Psi)^{-1}(x_t - \mu), \tag{6}$$

$$\mathbb{E}_{\mathbf{s}|x}[\mathbf{s}_t \mathbf{s}_t^T] = \text{Var}_{\mathbf{s}|x}[\mathbf{s}_t] + \mathbb{E}_{\mathbf{s}|x}[\mathbf{s}_t]\mathbb{E}_{\mathbf{s}|x}[\mathbf{s}_t]^T$$
$$= \Sigma_s - \Sigma_s^T W^T (W\Sigma_s W^T + \Psi)^{-1} W\Sigma_s + \mathbb{E}_{\mathbf{s}|x}[\mathbf{s}_t]\mathbb{E}_{\mathbf{s}|x}[\mathbf{s}_t]^T. \tag{7}$$

In the M-step, we update the parameter estimate to $\theta^{\text{new}}$ by maximizing $Q$ with respect to $W_i$, $\mu_i$, $\rho_i^2$, $i = 1{:}m$, and $\Sigma_s$. This is given by $\theta^{\text{new}} = \arg\max_\theta Q(\theta, \theta^{\text{old}})$, where

$$Q(\theta, \theta^{\text{old}}) = \tfrac{1}{d} \sum_{t=1}^{d} \int p(s_t|x_t, \theta^{\text{old}}) \log p(x_t, s_t|\theta) ds_t.$$

Due to the model structure, $Q$ can be maximized with respect to each parameter separately. To enforce the orthogonality of $W_i$, we bring a symmetric matrix $\Lambda_i$ of Lagrange multipliers and add the constraint term $\text{tr}(\Lambda_i(W_i^T W_i - I))$ to the objective function. Setting the derivatives of the modified objective to zero, we obtain the following update equations:

$$\mu_i^{\text{new}} = \tfrac{1}{d} \sum_t x_{it}, \tag{8}$$

$$W_i^{\text{new}} = A_i(A_i^T A_i)^{-1/2}, \quad A_i = \tfrac{1}{2}\left(\sum_t (x_{it} - \mu_i^{\text{new}})\mathbb{E}_{\mathbf{s}|x}[\mathbf{s}_t]^T\right), \tag{9}$$

$$\rho_i^{2\,\text{new}} = \tfrac{1}{dv}\sum_t \left(\|x_{it} - \mu_i^{\text{new}}\|^2 - 2(x_{it} - \mu_i^{\text{new}})^T W_i^{\text{new}}\mathbb{E}_{\mathbf{s}|x}[\mathbf{s}_t] + \text{tr}(\mathbb{E}_{\mathbf{s}|x}[\mathbf{s}_t \mathbf{s}_t^T])\right), \tag{10}$$

$$\Sigma_s^{\text{new}} = \tfrac{1}{d}\sum_t(\mathbb{E}_{\mathbf{s}|x}[\mathbf{s}_t \mathbf{s}_t^T]). \tag{11}$$

The orthonormal constraint $W_i^T W_i = I_k$ in SRM is similar to that of PCA. In general, there is no reason to believe that key brain response patterns are orthogonal. So, the orthonormal bases found via SRM are a computational tool to aid statistical analysis within an ROI. From a computational viewpoint, orthogonality has the advantage of robustness and preserving temporal geometry.

## 3.2 Connections with related methods

For one subject, SRM is similar to a variant of pPCA [15] that imposes an orthogonality constraint on the loading matrix. pPCA yields an orthogonal loading matrix. However, due to the increase in model complexity to handle multiple datasets, SRM has an explicit constraint of orthogonal loading matrices. Topographic Factor Analysis (TFA) [16] is a factor model using a topographic basis composed of spherical Gaussians with different centers and widths. This choice of basis is constraining but since each factor is an "blob" in the brain it has the advantage of providing a simple spatial interpretation. Hyperalignment (HA) [10], learns a shared representational by rotating subjects' time series responses to maximize inter-subject time series correlation. The formulation in [10] is based on problem (2) with $k = v$ and $W_i$ a $v \times v$ orthogonal matrix (Sup. Mat.). So this method does not directly reduce the dimension of the feature space, nor does it directly extend to this case (see Fig. 1). Although dimensionality reduction can be done posthoc using PCA, [10] shows that this doesn't lead to performance improvement. In contrast, we show in §4 that selecting $k \ll v$ can improve the performance of SRM beyond that attained by HA. The GICA, IVA algorithms [17] do not assume time-synchronized stimulus and hence concatenate data along the time dimension (implying spatial consistency) and learn spatial independent components. We use the assumption of a time-synchronized stimulus for anchoring the shared response to overcome a spatial mismatch in functional topographies. Finally, SRM can be regarded as a refinement of the concept of hyperalignment [10] cast into a probabilistic framework. The HA approach has connections with regularized CCA [18]. Additional details of these connections and connections with Canonical Correlation Analysis (CCA) [19], ridge regression, Independent Component Analysis (ICA) [20], regularized Hyperalignment [18] are discussed in the supplementary material.

## 4 Experiments

We assess the performance and robustness of SRM using fMRI datasets (Table 1) collected using different MRI machines, subjects, and preprocessing pipelines. The *sherlock* dataset was collected

| Dataset | Subjs | TRs (s/TR) | Region of interest (ROI) | Voxels |
|---|---|---|---|---|
| sherlock (audio-visual movie) [21] | 16 | 1976 (2) | posterior medial cortex (PMC) [22] | 813 |
| raider (audio-visual movie) [10] | 10 | 2203 (3) | ventral temporal cortex (VT) [23] | 500/H |
| forrest (audio movie) [24] | 18 | 3599 (2) | planum temporale (PT) [25] | 1300/H |
| audiobook (narrated story) [26] | 40 | 449 (2) | default mode network (DMN) [27] | 2500/H |

Table 1: fMRI datasets are shown in the left four columns, and the ROIs are shown in right two columns. The ROIs vary in functional from visual, language, memory, to mental states. H stands for hemisphere.

while subjects watched an episode of the BBC TV series "Sherlock" (66 mins).The *raider* dataset was collected while subjects viewed the movie "Raiders of the Lost Ark" (110 mins) and a series of still images (7 categories, 8 runs). The *forrest* dataset was collected while subjects listened to an auditory version of the film "Forrest Gump" (120 mins). The *audiobook* dataset was collected while subjects listened to a narrated story (15 mins) with two possible interpretations. Half of the subjects had a prior context favoring one interpretation, the other half had a prior context favoring the other interpretation. Post scanning questionnaires showed no difference in comprehension but a significant difference in interpretations between groups.

**Experiment 1: SRM and spatial smoothing**. We first use spatial smoothing to determine if we can detect a shared response in PMC for the *sherlock* dataset. The subjects are randomly partitioned into two equal sized groups, the data for each group is averaged, we calculate the Pearson correlation over voxels between these averaged responses for each time, then average these correlations over time. This is a measure of similarity of the sequence of brain maps in the two average responses. We repeat this for five random subject divisions and average the results. If there is a shared response, we expect a positive average correlation between the groups, but if functional topographies differ significantly across subjects, this correlation may be small. If the result not distinct from zero, a shared response is not detected. The computation yields the benchmark value $0.26 \pm 0.006$ shown as the purple bar in the right plot in Fig. 3. This is support for a shared response in PMC, but we posit that the subject's functional topographies in PMC are misaligned. To test this, we use a Gaussian filter, with width at half height of 3, 4, 5 and 6mm, to spatially smooth each subject's fMRI data, then recalculate the average Pearson correlation as described above. The results, shown as blue bars in Fig. 3, indicate higher correlations with greater spatial smoothing. This indicates greater average correlation of the responses at lower spatial frequencies, suggesting a fine scale mismatch of functional topographies across subjects.

We now test the robustness of SRM using the unsmoothed data. The subjects are randomly partitioned into two equal sized groups. The data in each group is divided in time into two halves, and the same half in each group is used to learn a shared response model for the group. The independently obtained group templates $S_1, S_2$, are then registered using a $k \times k$ orthogonal matrix $Q$ (method outlined in §2). For each group, the second half of the data is projected to feature space using the subject-specific bases and averaged. Then the Pearson correlation over features is calculated between the group averaged shared responses, and averaged over time. This is repeated using the other the halves of the subject's data for training and the results are averaged. The average results over 5 random subject divisions are report as the green bars in Fig. 3. With $k = 813$ there is no reduction of dimension and SRM achieves a correlation equivalent to 6mm spatial smoothing. This strong average correlation between groups, suggests some form of shared response. As expected, if the dimension of the feature space $k$ is reduced, the correlation increases. A smaller value of $k$, forces

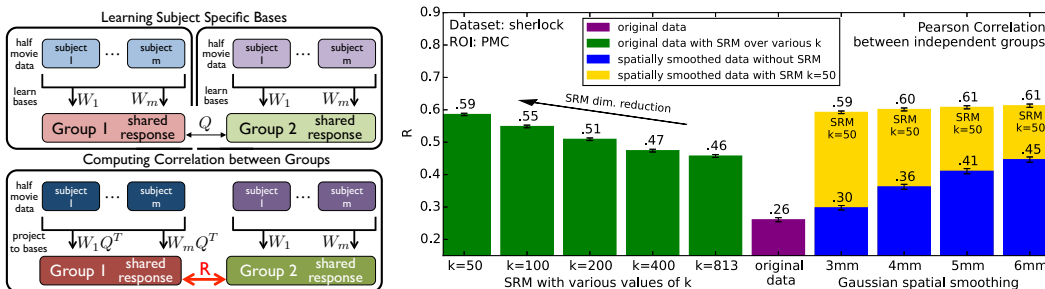

Figure 3: Experiment 1. Left: Learn using half of the data, then compute between group correlation on other half. Right: Pearson correlation after spatial smoothing, and SRM with various $k$. Error bars: $\pm 1$ stand. error.

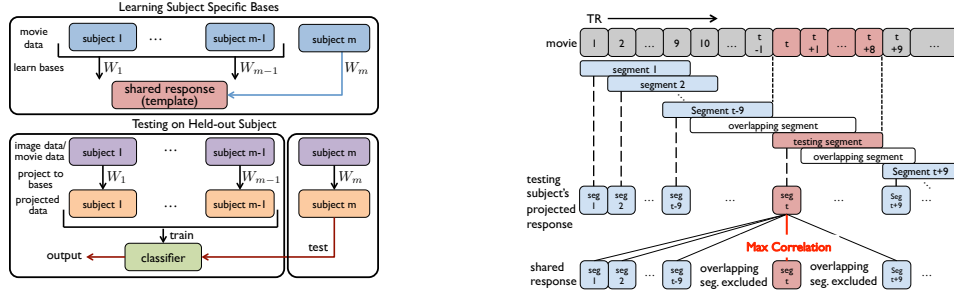

Figure 4: Left: Experiment 2. Learn subject specific bases. Test on held out subject and data. Right: Experiment 2. Time segment matching by correlating with 9 TR segments in the shared response.

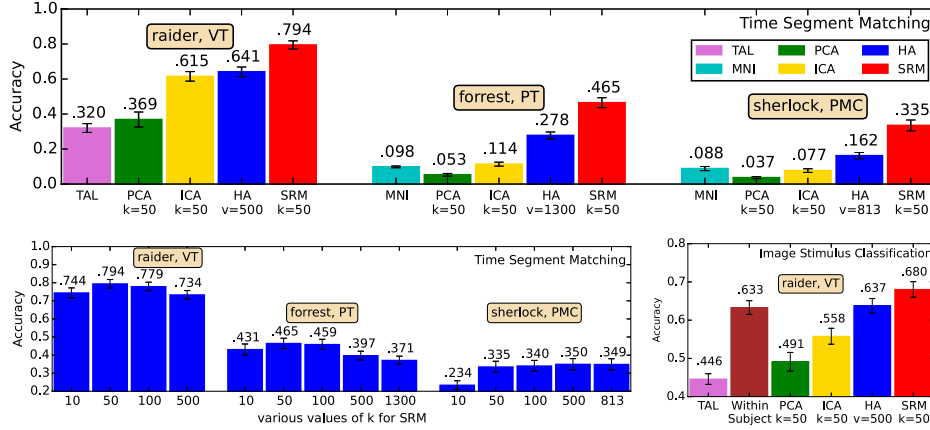

Figure 5: Experiment 2. Top: Comparison of 18s time segment classification on three datasets using distinct ROIs. Bottom: (Left) SRM time segment classification accuracy vs $k$. (Right) Learn bases from movie response, classify stimulus category using still image response. For *raider* and *forrest*, we conduct experiment on ROI in each hemisphere separately and then average the results. For *sherlock*, we conduct experiment over whole PMC. The TAL results for the *raider* dataset are from [10]. Error bars: $\pm 1$ stand. error.

SRM to focus on shared features yielding the best data representation and gives greater noise rejection. Learning 50 features achieves a 33% higher average correlation in feature space than is achieved by 6mm spatial smoothing in voxel space. A commensurate improvement occurs when SRM is applied to the spatially smoothed data.

**Experiment 2 : Time segment matching and image classification.** We test if the shared response estimated by SRM generalizes to new subjects and new data using versions of two experiments from [10] (unlike in [10], here the held out subject is not included in learning phase). The first experiment tests if an 18s time segment from a held-out subject's new data can be located in the corresponding new data of the training subjects. A shared response and subject specific bases are learned using half of the data, and the held out subject's basis is estimated using the shared response as a template. Then a random 18s test segment from the unused half of the held out subject's data is projected onto the subject's basis, and we locate the 18s segment in the averaged shared response of the other subject's new data that is maximally correlated with the test segment (see Fig. 4). The held out subject's test segment is correctly located (matched) if its correlation with the average shared response at the same time point is the highest; segments overlapping with the test segment are excluded. We record the average accuracy and standard error by two-fold cross-validation over the data halves and leave-one-out over subjects. The results using three different fMRI datasets with distinct ROIs are shown in the top plot of Fig. 5. The accuracy is compared using: anatomical alignment (MNI [4], Talairach (TAL) [1]); standard PCA, and ICA feature selection (FastICA implementation [20]); the Hyperalignment (HA) method [10]; and SRM. PCA and ICA are directly applied on joint data matrix $X^T = [X_1^T \ldots X_m^T]$ for learning $W$ and $S$, where $X \approx WS$ and $W^T = [W_1^T \ldots W_m^T]$. SRM demonstrates the best matching of the estimated shared temporal features of the methods tested. This suggests that the learned shared response is more informative of the shared brain state trajectory at an 18s time scale. Moreover, the experiment verifies generalization of the estimated shared features to subjects not included in the training phase and new (but similar) data collected during the other half of the movie stimulus. Since we expect accuracy to improve as the time segment is lengthened,

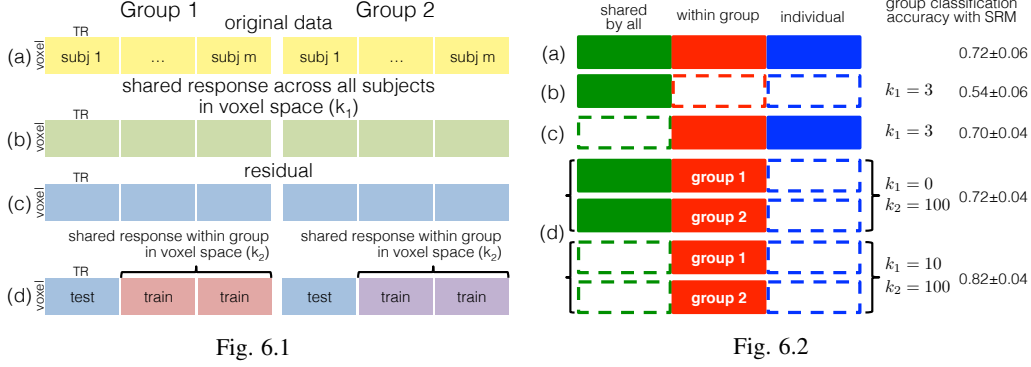

Fig. 6.1                                                    Fig. 6.2

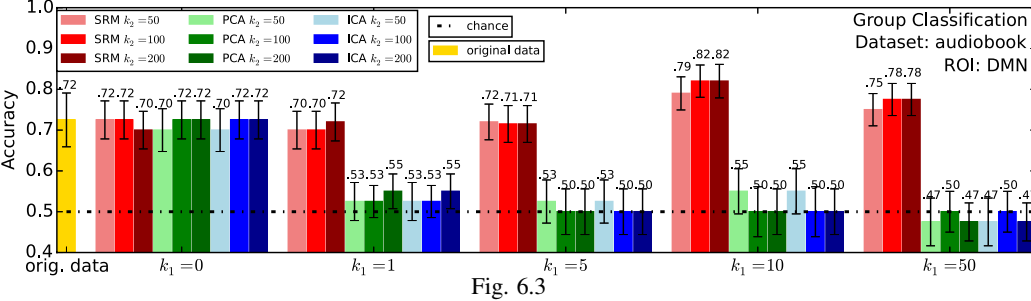

Fig. 6.3

Figure 6: Experiment 3. Fig. 6.1: Experimental procedure. Fig 6.2: Data components (left) and group classification performance with SRM (right) in different steps of the procedure. Fig. 6.3: Group classification on *audiobook* dataset in DMN before and after removing an estimated shared response for various values of $k_1$ and $k_2$ with SRM, PCA and ICA. Error bars: $\pm 1$ stand. error.

what is important is the relative accuracy of the compared methods. The method in (1) can be viewed as non-probabilistic SRM. In this experiment, it performs worse than SRM but better than the other compared methods. The effect of the number of features used in SRM is shown in Fig. 5, lower left. This can be used to select $k$. A similar test on the number of features used in PCA and ICA indicates lower performance than SRM (results not shown).

We now use the image viewing data and the movie data from the *raider* dataset to test the generalizability of a learned shared response to a held-out subject and new data under a very distinct stimulus. The *raider* movie data is used to learn a shared response model, while excluding a held-out subject. The held-out subject's basis is estimated by matching its movie response data to the estimated shared response. The effectiveness of the learned bases is then tested using the image viewing dataset [10]. After projecting the image data using the subject bases to feature space, an SVM classifier is trained and the average classifier accuracy and standard error is recorded by leave-one-out across subject testing. The results, lower right plot in Fig. 5, support the effectiveness of SRM in generalizing to a new subject and a distinct new stimulus. Under SRM, the image stimuli can be slightly more accurately identified using other subjects' data for training than using a subject's own data, indicating that the learned shared response is informative of image category.

**Experiment 3: Differentiating between groups.** Now consider the *audiobook* dataset and the DMN ROI. If subjects are given group labels according to the two prior contexts, a linear SVM classifier trained on labeled voxel space data and tested on the voxel space data of held out subjects, can distinguish the two groups at an above chance level. This is shown as the leftmost bar in the bottom figure of Fig. 6.3. This is consistent with previous similar studies [28].

We test if SRM can distinguish the two subject groups with a higher rate of success. To do so we use the procedure outlined in rows of Fig. 6.1. We first use the original data $X_{1:m}^{g1}, X_{1:m}^{g2}$ of all subjects (Fig. 6.1 (a)) to learn a $k_1$-dimensional shared response $S^{all}$ and subject bases $W_{gj,1:m}^{all}$. This shared response is then mapped to voxel space using each subject's learned topography (Fig. 6.1 (b)) and subtracted from the subject's data to form the residual response $X_i^{gj} - W_{gj,i}^{all} S^{all}$ for subject $i$ in group $j$ (Fig. 6.1 (c)). Leaving out one subject from each group, we use two within-group applications of SRM to find $k_2$-dimensional within-group shared responses $S^{g1}, S^{g2}$, and subject bases $W_{1:m}^{g1}, W_{1:m}^{g2}$

for the residual response. These are mapped into voxel space $W_i^{gj} S^{gj}$ for each subject (Fig. 6.1 (d)). The first application of SRM yields an estimate of the response shared by all subjects. This is used to form the residual response. The subsequent within-group applications of SRM to the residual give estimates of the within-group shared residual response. Both applications of SRM seek to remove components of the original response that are uninformative of group membership. Finally, a linear SVM classifier is trained using the voxel space group-labeled data, and tested on the voxel space data of held out subjects. The results are shown as the red bars in Fig. 6.3. When using $k_1 = 10$ and $k_2 = 100$, we observe significant improvement in distinguishing the groups.

One can visualize why this works using the cartoon in Fig. 6.2 showing the data for one subject modeled as the sum of three components: the response shared by all subjects, the response shared by subjects in the same group after the response shared by all subjects is removed, and a final residual term called the individual response (Fig. 6.2(a)). We first identify the response shared by all subjects (Fig. 6.2(b)); subtracting this from the subject response gives the residual (Fig. 6.2(c)). The second within-group application of SRM removes the individual response (Fig. 6.2(d)). By tuning $k_1$ in the first application of SRM and tuning $k_2$ in the second application of SRM, we estimate and remove the uninformative components while keeping the informative component.

Classification using the estimated shared response ($k_1 \leq 10$) results in accuracy around chance (Fig. 6.2(b)), indicating that it is uninformative for distinguishing the groups. The classification accuracy using the residual response is statistically equivalent to using the original data (Fig. 6.2(c)), indicating that only removing the response shared by all subjects is insufficient for improvement. The classification accuracy that results by not removing the shared response ($k_1 = 0$) and only applying within-group SRM (Fig. 6.2(d)), is also statistically equivalent to using the original data. This indicates that only removing the individual response is also insufficient for improvement. By combining both applications of SRM we remove both the response shared by all subjects and the individual responses, keeping only the responses shared within groups. For $k_1 = 10, k_2 = 100$, this leads to significant improvement in performance (Fig. 6.2(d) and Fig. 6.3).

We performed the same experiment using PCA and ICA (Fig. 6.3). In this case, after removing the estimated shared response ($k_1 \geq 1$) group identification quickly drops to chance since the shared response is informative of group difference (around 70% accuracy for distinguishing the groups (Sup. Mat.)). A detailed comparison of all three methods on the different steps of the procedure is given in the supplementary material.

## 5   Discussion and Conclusion

The vast majority of fMRI studies require aggregation of data across individuals. By identifying shared responses between the brains of different individuals, our model enhances fMRI analyses that use aggregated data to evaluate cognitive states. A key attribute of SRM is its built-in dimensionality reduction leading to a reduced-dimension shared feature space. We have shown that by tuning this dimensionality, the data-driven aggregation achieved by SRM demonstrates higher sensitivity in distinguishing multivariate functional responses across cognitive states. This was shown across a variety of datasets and anatomical brain regions of interest. This also opens the door for the identification of shared and individual responses. The identification of shared responses after SRM is of great interest, as it allows us to assess the degree to which functional topography is shared across subjects. Furthermore, the SRM allows the detection of group specific responses. This was demonstrated by removing an estimated shared response to increase sensitivity in detecting group differences. We posit that this technique can be adapted to examine an array of situations where group differences are the key experimental variable. The method can facilitate studies of how neural representations are influenced by cognitive manipulations or by factors such as genetics, clinical disorders, and development.

Successful decoding of a particular cognitive state (such as a stimulus category) in a given brain area provides evidence that information relevant to that cognitive state is present in the neural activity of that brain area. Conducting such analyses in locations spanning the brain, e.g., using a searchlight approach, can facilitate the discovery of information pathways. In addition, comparison of decoding accuracies between searchlights can suggest what kind of information is present and where it is concentrated in the brain. SRM provides a more sensitive method for conducting such investigations. This may also have direct application in designing better noninvasive brain-computer interfaces [29].

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
