[Supplementary Material]

# A Reduced-Dimension fMRI Shared Response Model Supplementary Material

**Po-Hsuan Chen**[1]**, Janice Chen**[2]**, Yaara Yeshurun**[2]**,**
**Uri Hasson**[2]**, James V. Haxby**[3]**, Peter J. Ramadge**[1]
[1]Department of Electrical Engineering, Princeton University
[2]Princeton Neuroscience Institute and Department of Psychology, Princeton University
[3]Department of Psychological and Brain Sciences
and Center for Cognitive Neuroscience, Dartmouth College

## S.1  Notation

| Variable | Description |
|---:|---|
| $v$ | number of voxels |
| $i$ | index for subject, $i \in \{1, \ldots, m\}$ |
| $t$ | index for TR, $t \in \{1, \ldots, d\}$ |
| $q$ | index for feature, $q \in \{1, \ldots, k\}$ |
| $\mathbf{x}_{it}$ | random variable for $t$-th observation from subject $i$, taking values in $\mathbb{R}^v$ |
| $x_{it}$ | $t$-th observation from subject $i$, $x_{it} \in \mathbb{R}^v$ |
| $X_i$ | observations from subject $i$, $X_i \in \mathbb{R}^{v \times d}$ |
| $\mathbf{x}_t$ | random variable for observation of $t$-th observations from all subjects, taking values in $\mathbb{R}^{mv}$ |
| $x_t$ | $x_t^T = [x_{1t}{}^T \ldots x_{it}{}^T]$, $x_t \in \mathbb{R}^{mv}$ <br> concatenated observation of $t$-th observations from all subjects |
| $X$ | $X^T = [X_1{}^T \ldots X_i{}^T]$, $X \in \mathbb{R}^{mv \times d}$ <br> concatenated observations from all subjects |
| $\mathbf{s}_t$ | random variable for shared response of $t$-th observations, taking value in $\mathbb{R}^k$ |
| $s_t$ | estimated shared response of $t$-th observations, $s_t \in \mathbb{R}^v$ |
| $S$ | estimated shared response , $S \in \mathbb{R}^{k \times d}$ |
| $\mu_i$ | mean observation from subject $i$, $\mu_i \in \mathbb{R}^v$ |
| $\mu$ | $\mu^T = [\mu_1^T \ldots \mu_M^T]$, $\mu \in \mathbb{R}^{mv}$ <br> concatenated mean observation from all subjects |
| $W_i$ | loading matrix for subject $i$, $W_i \in \mathbb{R}^{v \times k}$ |
| $W$ | $W^T = [W_1^T \ldots W_m^T]$, $W \in \mathbb{R}^{mv \times k}$ <br> concatenated loading matrix for all subjects |
| $\Sigma_s$ | covariance for shared response $\mathbf{s}_t$, $\Sigma_s \in \mathbb{R}^{k \times k}$ |
| $\rho_i^2 I_v$ | isotropic covariance for conditional distribution of $\mathbf{x}_{it}$ |
| $\Psi$ | $\Psi = \text{diag}(\rho_1^2 I, \ldots, \rho_m^2 I)$, $\Psi \in \mathbb{R}^{mv \times mv}$ <br> joint covariance for condition distribution of $\mathbf{x}_t$ |

## S.2 Identity between two approaches when $k = v$

$$\min_{W_i,S} \quad \sum_i \|X_i - W_i S\|_F^2 \qquad (1)$$
$$\text{s.t.} \quad W_i^T W_i = I_k,$$

$$\min_{W_i,S} \quad \sum_i \|W_i^T X_i - S\|_F^2 \qquad (2)$$
$$\text{s.t.} \quad W_i^T W_i = I_k,$$

We show the identity between (1) and (2) when $k = v$, square orthogonal matrices $W_i$.

$$\begin{aligned}
&\|W_i^T X_i - S\|_F^2 \\
=&\text{tr}\big((W_i^T X_i - S)^T (W_i^T X_i - S)\big) \\
=&\text{tr}\big((W_i^T X_i - S)^T W_i^T W_i (W_i^T X_i - S)\big) \\
=&\text{tr}\big((X_i - W_i S)^T (X_i - W_i S)\big) \\
=&\|X_i - W_i S\|_F^2
\end{aligned} \qquad (3)$$

When $W_i$ is a narrow matrix with orthonormal columns, (3) is false, because $W_i W_i^T \neq I$.

## S.3 Difference between two approaches

We show this by analyzing (1) and (2). $X_i$ can be decomposed as $X_i^W + X_i^{W\perp}$, where $X_i^W = W_i W_i^T X_i$ is the part of $X_i$ in the span of $W_i$ and $X_i^{W\perp} = W_i^\perp {W_i^\perp}^T X_i$ is the part of $X_i$ in the orthogonal complement of span of $W_i$. By expanding (1) and (2), we get:

$$\|W_i^T X_i - S\|_F^2 = \text{tr}({X_i^W}^T X_i^W) - 2\text{tr}({X_i^W}^T W_i S) + \text{tr}(S^T S) \qquad (4)$$

$$\|X_i - W_i S\|_F^2 = \text{tr}({X_i^W}^T X_i^W) + \text{tr}({X_i^{W\perp}}^T X_i^{W\perp}) - 2\text{tr}({X_i^W}^T W_i S) + \text{tr}(S^T S) \qquad (5)$$

For (5), since $\text{tr}({X_i^W}^T X_i^W) + \text{tr}({X_i^{W\perp}}^T X_i^{W\perp}) = \text{tr}(X_i^T X_i)$, it's trying to find $W_i$ maximizing $\text{tr}({X_i^W}^T W_i S)$, maximizing the correlation between transformed observation $W_i^T X_i$ and the shared response $S$. However, for (4), there's a conflict between the first and second terms. The first term is minimizing the variance of projected data, while the second term is maximizing the variance of projected data $X_i^W$ with shared response. Due to this conflict, (2) is prone to find an uninformative basis $W_i$ which doesn't capture the variance in observations. This is verified in Fig.1 of the main paper, which shows plots of the value of the training objective and the test accuracy of a stimulus classification experiment versus iteration count (*raider* dataset, see Sec.4).

### S.3.1 SRM is adaptively aggregating data with different estimated noise level

We note that (1) implicitly assumes subjects having identical noise level. This is reflected by the update equation for $S$ taking a uniform average of the transformed data. In SRM, if instead we set the estimated value of $\rho_i^2$ to be $\kappa_i^2 \lambda^z$ for $0 < \lambda < 1$, $i = 1:m$, and let $z \to \infty$, then the shared response becomes a weighted average of the transformed data in which subjects with less noise are weighted more:

$$\lim_{z \to \infty} \mathbb{E}_{s|x}[S] = \lim_{z \to \infty} \sum_i \text{diag}\left\{\frac{\kappa_i^{-2}\lambda^{-z}}{\sum_j \kappa_j^{-2}\lambda^{-z} + \sigma_i^{-1}}\right\} W_i^T X_i = \sum_i \frac{1/\kappa_i^2}{\sum_j (1/\kappa_j^2)} W_i^T X_i.$$

## S.4 Connections between SRM and related methods

### S.4.1 SRM and CCA solutions as different parameterization of two subjects pCCA likelihood

Bach and Jordan [1] proposed probabilistic CCA (pCCA) using similar approach as Tipping and Bishop used in probabilistic PCA (pPCA) [2]. They propose a probabilistic model of pPCA and prove that maximum likelihood estimation leads to the canonical correlation directions in two subjects case. Probabilistic CCA is proposed as follows:

$$\mathbf{z} \sim \mathcal{N}(0, I), \mathbf{z} \in \mathbb{R}^k, k \leq v$$
$$\mathbf{x}_1|\mathbf{z} \sim \mathcal{N}(W_1 \mathbf{z} + \mu_1, \Psi_1)$$
$$\mathbf{x}_2|\mathbf{z} \sim \mathcal{N}(W_2 \mathbf{z} + \mu_2, \Psi_2),$$

where $x_1$ and $x_2$ take values in $\mathbb{R}^v$. with maximum likelihood estimates:

$$\widehat{W}_1 = \widetilde{\Sigma}_{11}\widehat{U}_1\widehat{M}_1$$
$$\widehat{W}_2 = \widetilde{\Sigma}_{22}\widehat{U}_2\widehat{M}_2$$
$$\widehat{\Psi}_1 = \widetilde{\Sigma}_{11} - \widehat{W}_1\widehat{W}_1^T$$
$$\widehat{\Psi}_2 = \widetilde{\Sigma}_{22} - \widehat{W}_2\widehat{W}_2^T$$
$$\hat{\mu}_1 = \tilde{u}_1$$
$$\hat{\mu}_2 = \tilde{u}_2$$

where $\widehat{U}_i = \widetilde{\Sigma}_{ii}^{-\frac{1}{2}}\widehat{V}_i$, $\widehat{V}_1\widehat{P}\widehat{V}_2^T$ is a SVD of $\widetilde{\Sigma}_{11}^{-\frac{1}{2}}\widetilde{\Sigma}_{12}\widetilde{\Sigma}_{22}^{-\frac{1}{2}}$. $\widehat{P}$ can be factorized as $\widehat{M}_1\widehat{M}_2$. $\widehat{U}_i$ is the transformation matrix for dataset $i$ in CCA solutions. The corresponding log-likelihood value is

$$\mathcal{L}\big|_{\widehat{W},\widehat{\Psi},\widehat{\mu}} = -\frac{2vd}{2}\log 2\pi e - \frac{d}{2}\log|\widetilde{\Sigma}_x|$$

Next, following similar approach as in [1] we can show that a different mode of maximum likelihood estimates of pCCA leads to close relation with SRM. We derive this mode by taking derivative of the log-likelihood but using different parameterization as the pCCA. The maximum likelihood estimates are:

$$\overline{W}_1 = \overline{U}_1\overline{M}_1$$
$$\overline{W}_2 = \overline{U}_2\overline{M}_2$$
$$\overline{\Psi}_1 = \widetilde{\Sigma}_{11} - \overline{W}_1\overline{W}_1^T$$
$$\overline{\Psi}_2 = \widetilde{\Sigma}_{22} - \overline{W}_2\overline{W}_2^T$$
$$\overline{\mu}_1 = \tilde{u}_1$$
$$\overline{\mu}_2 = \tilde{u}_2$$

where $\overline{U}_1\overline{P}\,\overline{U}_2^T$ is a SVD of $\widetilde{\Sigma}_{12}$. $\overline{P}$ can be factorized as $\overline{M}_1\overline{M}_2$. $\overline{U}_i$ is the orthogonal transformation matrix for dataset $i$ in SRM solutions. The corresponding log-likelihood value is

$$\mathcal{L}\big|_{\overline{W},\overline{\Psi},\overline{\mu}} = -\frac{2vd}{2}\log 2\pi e - \frac{d}{2}\log|\widetilde{\Sigma}_x|$$

which is equal to the log-likelihood derived in pCCA. This shows that SRM and CCA solutions in two subjects case are different parameterization of the same pCCA likelihood.

### S.4.2  Connections between SRM and ridge regression

SRM is related to ridge regression. We make this connection by showing that single subject SRM is connected with ridge regression with an orthogonality constraint on the loading matrix. Assume $\mathbf{s}_t$ is sampled from $\mathcal{N}(0, \gamma^2 I)$ with $\gamma^2$ known, and that $\Sigma_{x'_m} = I$. When $M = 1$, MAP estimation of $W_i$ and $s_t$, $t = 1{:}T$, estimates a mode of the log posterior distribution $\sum_t \log p(s_t|x_{it})$:

$$\max \sum_t \big(\log p(x_{it}|s_t) + \log p(s_t)\big) \equiv \min \sum_t \big(\|x_{it} - W_i s_t\|_F^2 + \gamma^{-2}\|s_t\|_2^2\big).$$

This is ridge regression for $\mathbf{s}_t$ given $W_i$, and least squares regression for $W_i$ (with an orthogonality constraint), given $\mathbf{s}_t$, $t = 1:T$. In the multi-subject case, MAP estimation of $W_i$ and $\mathbf{s}_t$, will be similar but with a block-wise orthonormal structure in $W$:

$$\max \sum_t \sum_i \big(\log p(x_{it}|s_t) + \log p(s_t)\big) \equiv \min \sum_t \big(\|x_t - W s_t\|_F^2 + \gamma^{-2}\|s_t\|_2^2\big).$$

### S.4.3  Connections between SRM and Hyperalignment

We show that Hyperalignment[3] is equivalent to (2) when $k = v$. Following is the formulation of Hyperalignment, note that $X_i \in \mathbb{R}^{v \times d}$ here is the transpose of the notation used in [3].

$$\begin{aligned}\min_{R_i} \quad & \sum_{i<j}\|X_i^T R_i - X_j^T R_j\|_F^2 \\ \text{s.t.} \quad & R_i^T R_i = R_i R_i^T = I_v,\end{aligned} \tag{6}$$

From the equality $\sum_{i<j}\|X_i^T R_i - X_j^T R_j\|_F^2 = \sum_i \|X_i^T R_i - G\|_F^2$ [4], where $G = {}^{1}/{m}\sum_i X_i^T R_i$, by letting $W_i = R_i$ and $S = G^T$, we get identical formulation as (2) when $k = v$.

### S.4.4 Connections between SRM and regularized Hyperalignment

Lastly, we show the difference between SRM and regularized HA (rHA) in [5]. rHA makes a connection between HA and CCA [6] using a ridge CCA formulation [7]. We show rHA on the left and a matching formulation in SRM notation on the right:

$$
\begin{aligned}
&\min \ \sum_{i<j} \|X_i^T R_i - X_j^T R_j\|_F^2 \\
&\text{s.t.} \ \ R_i^T((1-\alpha)X_i X_i^T + \alpha I)R_i = I
\end{aligned}
\quad \equiv \quad
\begin{aligned}
&\min \ \sum_i \|W_i^T X_i - S\|_F^2 \\
&\text{s.t.} \ \ W_i^T((1-\alpha)X_i X_i^T + \alpha I)W_i = I.
\end{aligned}
$$

rHA introduces a parameter $\alpha$ bridging the HA constraint $R_i^T R_i = I$ and the CCA constraint $R_i^T X_i^T X_i R_i = I$. rHA becomes standard HA when $\alpha \to 1$ and CCA when $\alpha \to 0$. In contrast to the regularization on the loading matrices $W_i$ imposed by rHA, SRM introduces regularization on the shared randomness $\mathbf{s}_t$.

### S.4.5 Connections between SRM and standard ICA

standard ICA [8] is a factor model that tries to find linear representation of data so that the components are statistically independent, or as independent as possible. Using our notation, given data $X$,

$$x = Ws,$$

where $s$ is the independent components. There are two main differences between ICA and SRM. First, ICA isn't designed for multiple datasets. Although there are multiple datasets extension of ICA, such as GICA, IVA, the difference is discussed in the main paper. Second, ICA doesn't have the notion of "shared response". It's maximizing statistical independence, but this doesn't necessarily lead to shared components. This is examined in experiment 3 of the main paper.

## S.5  Experiment 3

Fig. 1.1 Experiment procedure.

Fig. 1.2 On original data.

Fig. 1.3 On removed shared across all response.

Fig. 1.4 On residual after removed shared across all response.

Fig. 1.5 After removed shared across all response and train on shared within group response.

Figure 1: Experiment 3. Fig. 1.1: Experiment procedure. Fig. 1.2: On original data. Fig. 1.3: On removed shared across all response. Fig. 1.4: On residual after removed shared across all response. Fig. 1.5: After removed shared across all response and train on shared within group response.

In Fig. 1.2, we show the classification accuracy directly on original data. The above chance results indicates that the groups are distinguishable in DMN ROI.

Fig. 1.3 shows that the Fig. 1.1 (b) is uninformative for SRM with low $k_1$ but informative for SRM with large $k_1$, PCA and ICA. The shared by all subjects response is expected to be uninformative for distinguishing between groups. However, when we use large $k_1$, SRM starts to incorporate shared within group only response as shared by all subjects response, because we force it to identify a large subspace. This is demonstrated by the above chance performance with $k_1 = 100$. As for PCA and ICA, they identify components that lead to maximum variance and statistical independence respectively. It's doesn't guarantee that the shared response will be identified by these two methods. So PCA and ICA cant be relied upon to identify a shared response. The above chance accuracy suggests that it's indeed not identifying shared by all subjects response.

Fig. 1.4 shows that Fig. 1.1 (c) is uninformative for PCA and ICA, because the informative part of original data has been removed with Fig. 1.1 (b). For SRM, we observe consistent above chance performance in distinguishing groups. The performance is similar to on original data.

In Fig. 1.5, without removing shared response ($k_1 = 0$), we observe that all three methods are effective in distinguishing between two groups. However, this doesn't lead to better performance than on original data. This suggests removing individual response is insufficient for improvement. However, with proper selection of $k_1$ and $k_2$, we observe statistically significant improvement with $k_1 = 10$ and $k_2 = 100$. This shows that by removing both shared by all subjects response and individual response, the denoised data demonstrates better distinguishability for the groups.