[Reviews · NeurIPS 2015]

Submitted by Assigned_Reviewer_1

The manuscript "A Reduced-Dimension fMRI Shared Response Model" provides a model for correspondence across multi-subject fMRI datasets, related to the popular hyper-alignment model.

I appreciate the fact that the paper does a good effort to build a sound mathematical model behind hyper-alignment related algorithms and provide good validation. Given the popularity and promises of such techniques, this work in important. The manuscript is in general clear and well written, although some details about the validation could be added. The work is original: it does not claim to be a radical departure from the prior art, which often reflects a lack of knowledge of the literature, but explains well how it relates to previous work, and how it improves significantly upon it.

Below I give specific comments that I believe that authors can use to improve their paper.

To me, an important aspect of the paper is that it reframes the popular hyper alignement technique and improves upon. I think that the formulation of paper does not make it appear enough. The term "hyper alignment", that is well known to the fMRI world, should be used more, in particular in the abstract, introduction, and conclusion. I understand that the authors may fear that it could make their work look incremental, however, nothing is built from a void, and stressing the relationship to previous work is a strength, given that this contribution improves upon it.

I believe that experiment 1 does not show what the authors claim that it shows, in particular the sentence 'This supports the claim of a shared response with distinct functional topographies across subjects.'. Indeed, smoothing reduces the degrees of freedom of the noise. Thus is makes the variance of a comparison across subjects decrease. If there is common information across subjects, the correlation increases mechanically as long as the amplitude of the common information decreases less than the amplitude of the noise (as in the matched filter theorem). This can be shown with very simple simulations using random signals. Thus the increase of correlation across subjects is actually quite trivial. The problem is that smoothing and SRM are both transformations that shrink the volume of the accessible space. The smaller this volume, the more likely correlations are to be strong. In this regard, I do not think that experiment 1 is telling us much. I think that the limitation of correlation as a validation metric appears in the fact that the authors fully explore the logic of maximizing correlation. Indeed, it would be interesting to increase smoothing, and decrease k, until a peak in correlation is reached. Both of these cases would lead to exploring a very restricted signal space, and our intuition tells us that we would probably be throwing out the baby with the bath water. I think that the paragraphs on experiment 1 need to be significantly reworked and downplayed, as they do not provide much evidence that SRM is specific to common signal across subjects. One interesting variant of the experiment would however be to show that when varying SRM parameters, the peak of correlation is higher than when varying smoothing. This would show that SRM is a better filter than smoothing to extract common information.

In experiment 2, what was the smoothing used by for the spatial correspondence methods (Talairach and MNI)? Is an optimal in classification reached through varying this parameter?

In experiment 2 and 3, more details should be given on how PCA and ICA are used. Right now, it is not possible to reproduce the experiments. In particular, I don't understand the difference between the performances of ICA and PCA, given that 1) the learner used (SVM) is rotationally-invariant and 2) ICA and PCA only differ by a rotation.

As a side note, in experiment 3, the sub-space spanned by ICA and PCA are the same, thus it makes sense that they perform the same.

The authors have shown that the dimensionality reduction of their SRM method what a strong benefit to prediction accuracy. I am curious to know what dimensionality they used for the ICA and PCA approaches, and if the results of the ICA and the PCA could be improved by varying this dimensionality.
Summary: This manuscript tackles an important class of techniques for multi-subject fMRI related to hyper alignment, for which it contributes a sound probabilistic model and a variate. There is a strong effort to validate the method which is precious, although I believe that some validations have loopholes.

Submitted by Assigned_Reviewer_2

This paper proposes a model to analyze cross-subject fMRI images, where each subject's fMRI image X_i is modeled as W_i * S where W_i is a low-rank subspace, and S is the shared response across subjects. The model is slightly different from a common subspace model (or principal component analysis (PCA)) because the subspace W_i is different for each subject. The paper shows empirically that this new model works better than PCA and ICA (independent component analysis). Overall the paper looks interesting and the results look convincing, though the model is not clearly justified as a reasonable model for fMRI responses. The authors do not explain why their model is superior (e.g., why a "shared response" model makes more sense than a "shared subspace" model), which is less than satisfying.

A question about a notation: what does TR mean in this paper? It does not look like it is used properly.
Summary: Overall the paper looks interesting and the results look convincing, though the model is not clearly justified as a reasonable model for fMRI responses.

Submitted by Assigned_Reviewer_3

The paper #344, entitled 'A Reduced-Dimension fMRI Shared Response Model', presents a supervised multivariate learning framework for brain images. After revisiting the distinction between discriminative versus generative models in the context of high-dimensional linear models for functional imaging, the authors propose the so-called SRM framework, which takes into account the shared information across datasets acquired under the same condition in different subjects, to improve the estimation of time series. Experiments are performed on four different datasets where the SRM is compared to rather simple competitors (PCA, ICA, univariate models).

There are very good aspects in that paper. First, the proposed method is arguably simple, yet sensible and well described; it is shown to perform well, as expected. Second, the relatively complete validations are really appreciated. This is the only paper in my stack that presented reasonable validation experiments -- on several datasets.

The idea behind the SRM is actually not much different from a generalized canonical correlation analysis --or more exactly: multi-block PLS-- framework, that have already been defined in this field (see e.g. [0, 1, 2, 3]), even in more sophisticated settings [2]. Section 3.2 is thus dramatically incomplete, as the authors perform a rather biased selection of anterior work. These anterior contributions are neither acknowledged properly nor used as benchmark. What is the point in designing new methods if there is no serious comparison with the state of the art ?

Second, there is a blatant inconsistency about the issues of regional characterization. The authors start the paper by discussing the problem of registration on fMRI group studies, which is important indeed. However, they then rely on atlas-defined ROIs (that have no functional relevance) to perform their analysis. This sounds wrong: if one does not trust spatial alignment, one cannot make proper inference on atlas-defined structures.

Misc technical issues:

- Due to non-unicity, the resulting decomposition may not be thought of as providing intrinsic characteristics of the dataset under consideration. Without a clear statement on the characterization of the solution (or lack thereof), I think that the paper is not acceptable at NIPS.

- The authors introduce a probabilistic setting, which is novel if we compare the proposed approach to existing multi-subject dictionary learning works, but obviously useless, as it is never used in the experiments. To me there is no more here than rhetorical arguments.

- Finally the authors used orthogonality constraints on the components, which seems to be a rather arbitrary choice, possibly counter-productive. Could this be discussed ?

Experiments: I find experiment 1 a bit disappointing, as it gives rather non-specific/trivial results, that are well-known (see e.g. [4]). Experiment 2 is clearly more interesting, but it feels a bit strange that ICA/PCA are failing so dramatically. I would not conclude about it without a thorough comparison with e.g. [1]

[0] Friman O, Cedefamn J, Lundberg P, Borga M, Knutsson H. Detection of neural activity in functional MRI using canonical correlation analysis. Magn Reson Med. 2001 Feb;45(2):323-30.

[1] Varoquaux G, Sadaghiani S, Pinel P, Kleinschmidt A, Poline JB, Thirion B. A group model for stable multi-subject ICA on fMRI datasets. Neuroimage. 2010 May 15;51(1):288-99.

[2] Varoquaux G, Schwartz Y, Pinel P, Thirion B. Cohort-level brain mapping: learning cognitive atoms to single out specialized regions. Inf Process Med Imaging. 2013;23:438-49.

[3] Afshin-Pour B, Hossein-Zadeh GA, Strother SC, Soltanian-Zadeh H. Enhancing reproducibility of fMRI statistical maps using generalized canonical correlation analysis in NPAIRS framework. Neuroimage. 2012 May 1;60(4):1970-81.

[4] Zhang J, Anderson JR, Liang L, Pulapura SK, Gatewood L, Rottenberg DA, Strother SC. Evaluation and optimization of fMRI single-subject processing pipelines with NPAIRS and second-level CVA. Magn Reson Imaging. 2009 Feb;27(2):264-78.

Minor points ------------ - the abstract does not state a very specific problem. - l.16 anatomically aligned
Summary: The technical contribution is not impressive, but the framework is sound, well explained and well validated. This is an OK paper overall.

Submitted by Assigned_Reviewer_4

This paper presents a technique to extract common signals and individual topographies when subjects are scanned while being presented with the same stimuli. It is tested on three datasets and shown to be an efficient decomposition for classification.

The paper is clear and dense. I believe it is mostly original, similar techniques have been presented but the authors decomposition seem to outperform all previous solutions. The exploration of the technique over three datasets shows its potential and significance for functional brain imaging. The aspect that may lack discussion in some part of the paper is the choice of the dimensionality k.

Overall I thought that this was a very good and interesting paper.

Some questions/remarks:

section 2: l.117: this step seems to be simple OLS regression and could be refered to as it l.122: with different stimuli the generated W_i should be different - I am not sure what the authors have in mind ere

section 3:

l.149: the isotropic covariance is clearly a too strong assumption - done because of the dimensionality of the data - do the author know what is the likely result of that assumption? 3.1 EM: will converge towards a local min/max: what is the depency of the results when varying starting points?

section experiments: In general - how were the data pre-processed? Experiments 1 and 2 are clear - I have found experiment 3 harder to follow (removing the individual signal means that only the within group signal is used ?) but interesting.

For experiment 1: why not learn the common signal on all train data (group1 and group2) ?

Summary: This is a interesting paper that presents a decomposition technique to extract common temporal signals (and subject specific topographies) in fMRI multisubject data, and tests the technique on three different datasets.

Submitted by Assigned_Reviewer_5

(light review)

compare to gCCA or aCVA??

Line 237: specify that each correlation is between corresponding voxels from two different images.

At the end of experiment 3, Line 407 make one of the most important claims of the paper, but unfortunately I wasn't convinced on the basis of my admittedly light reading. I expect you're right, but please spell this out in more detail.

Summary: Simple, compelling model for an important problem. Generally well-explained, with informative experiments. Very nice supplementary material.

Author Feedback
Author rebuttal: We thank the reviewers for their questions and helpful comments.

AR1.1 Your point that correlation increases with greater constraints is clear. Our intent was to compare the effect of SRM with spatial smoothing, a standard method used in multi-subject fMRI analysis. Without dimensionality reduction SRM increases correlation to that achieved by 6mm spatial. In this case SRM is an invertible transformation, so no information has been lost. Dimensionality reduction will further increase correlation but we need to show that this is capturing truly shared information - this is the purpose of Exp 2 and 3. We will reword our explanation to avoid overstating conclusions. An additional aspect of Exp 1, is to investigate whether SRM is consistent across two independent groups under an identical stimulus.

AR1.2 Smoothing parameters in Exp 2 are 3mm for TAL and 2mm for MNI. These parameters were selected in the different labs' pipelines and were not necessarily tuned for optimal performance in our experiments. However, performance is fairly stable wrt this parameter.

AR1.3 We will include more details in how PCA and ICA are used in Exp 2 and 3, and discuss the distinction between their performances.

AR1.4 The prediction accuracy for ICA and PCA differs with k but for all k, ICA and PCA perform worse than SRM.

AR2, AR5: We thank the reviewers for pointing out refs [0-4]. We will study these and if possible include experimental comparisons. It will take time to digest these works but the connections will be explored. SRM is an extension of the HA approach, and prior work has compared regularized CCA and HA. We will point this out.

AR2.1 SRM does not rely on anatomical ROIs. In our experiments we use anatomic ROIs only in the role of selecting a potentially interesting subset of voxels. We used these particular ROIs because these were of interest to the labs that collected the data. But they are not critical to SRM; one could instead use a local searchlight or perhaps a functionally selected subset of voxels.

AR2.2 To keep the paper concise, we excluded discussion of a non-probabilistic version of SRM. It performs worse than SRM but is also better than the other compared methods. We will add a note to this effect.

AR2.3 Orthogonality should not be essential but in aids computation and seems to have an advantage in robustness. The downside is that the basis may not be easy to interpret. Our initial tests on dropping the constraint leads to decreased performance.

AR3, AR6:
The idea is that with a rich stimulus, we learn a W that generalizes to different stimuli. We evaluated the model using various datasets with rich stimuli, and showed the generalizability of W by testing on different stimuli or distinct parts of the same stimulus.

AR3:
1. k is chosen using cross-validation.
2. l.117 is Procrustes problem, different from OLS due to the orthogonality constraint.
3. The isotropic covariance assumption yields closed-form update equations.
4. SRM is robust over different initializations; results shown are averaged over multiple random initializations.
5. Minimal standard pre-processing steps include: motion correction, slice time correction, detrending, and coregistration to standard space (MNI/TAL). Additional smoothing was then applied.
6. We will improve the clarity of experiment 3.
7. The approach of learning on all training subjects is used in experiment 2. In experiment 1, the idea is to investigate whether SRM is consistent across two independent groups under an identical stimulus.

AR4:
It is called "shared response model" because we explicitly model shared response as latent variable. TR (Time of Repetition) is the time between successive excitation pulses.